# Effect of Mindfulness on the Stress–Recovery Balance in Professional Soccer Players during the Competitive Season

**Joaquín Holguín-Ramírez** [1], **Arnulfo Ramos-Jiménez** [2,]*, **José Trinidad Quezada-Chacón** [2], **Mónica Sofía Cervantes-Borunda** [1] **and Rosa Patricia Hernández-Torres** [1,]*

[1]  Facultad de Ciencias de la Cultura Física, Universidad Autónoma de Chihuahua, Chihuahua 31000, Mexico; joaquin.holguin.ram@gmail.com (J.H.-R.); mcervant@uach.mx (M.S.C.-B.)

[2]  Departamento de Ciencias de la Salud, Instituto de Ciencias Biomédicas, Universidad Autónoma de Ciudad Juárez, Chihuahua 32310, Mexico; jose.quezada@uacj.mx

*   Correspondence: aramos@uacj.mx (A.R.-J.); rhernant@uach.mx (R.P.H.-T.); Tel.: +52-656-167-9309 (A.R.-J.); +52-656-311-1355 (R.P.H.-T.)

**Abstract:** Professional athletes are subjected to constant stress that often leads them to exceed their capacities and lose their homeostasis, which without a proper recovery program can lead to injury, chronic fatigue, and overtraining. This work examines the effect of six weeks of a psychological intervention called Mindful Sports Performance Enhancement (MSPE), on the stress–recovery balance in professional soccer players during a competitive season. Methods: The RESTQ-76 Sport psychometric questionnaire and heart rate variability (HRV) were used as psychometric and physiological evaluation methods. Under a longitudinal case–control study and having complied with bioethical procedures, 42 professional soccer players (22 control without treatment and 20 experimental), age 17 ± 1 year, weight 63 ± 11 kg, and height 172 ± 7 cm, were analyzed. Results: RESTQ-76 Sport increased the stress–recovery balance and global recovery ($p < 0.5$), but decreased global stress. The stress–recovery balance values measured by the nonlinear indicators of the HRV: SD1, SD2, SS, and S:PS, were not modified. Conclusions: Six weeks of MSPE improves the stress–recovery balance in third-division professional soccer players during the competitive season, reduces stress, and increases recovery. These positive effects were not observed in the nonlinear indicators of the HRV: SD1, SD2, SS, and S:PS.

**Keywords:** physical exercise; sport psychology; stress disorder; complementary therapy; athletic performance

## 1. Introduction

Professional athletes have to deal with various forms of sport-specific stressors, such as training load, social, media, schoolwork studies, and competitive pressures, leading athletes to face high levels of physical, mental, and emotional stress. When these pressures are not accompanied by active recovery periods, athletes often overload their physical and psychological adaptation, presenting a high and chronic activation of the sympathetic nervous system [1,2] and developing overtraining syndrome, and in extreme cases, burnout [3–5]. These syndromes are associated with physical, mental, and emotional fatigue, decreased sports performance, infectious diseases, musculoskeletal injuries, depression, hostility, inattention, and personal and professional abandonment [3,6]. Given this problem, various psychological interventions have been applied to reduce stress through controlling thoughts, improving alertness, fixing images, sounds, meditation, and concentration [7]. However, several of these techniques are not validated or are difficult to replicate, especially in sports activities. In this

sense, Kaufman, Glass, and Pineau (2018) [8], based on the guidelines of Gardner and Moore (2007) [9], created Mindful Sports Performance Enhancement (MSPE), which is a well-standardized psychological intervention, proposed to reduce stress and anxiety and improve the alertness of athletes, also on the validation process. To avoid cognitive disorders and negative emotions, MSPE invites the athlete to practice the following three life aspects: intention, attention, and attitude, thereby promoting the acceptance of thoughts and feelings without prejudice, allowing them to be focused on recovery [10]. Further, MSPE allows the athlete to obtain the ability to reevaluate competitive challenges and regulate their stress response during competition [8].

Low-quality studies have shown the effectiveness of mindfulness in reducing stress levels and improving sports performance in healthy people [11]. Kaufman et al. (2009), in a longitudinal study made up of archers and golfers, showed that the MSPE program decreases anxiety, fixation on perfection, and disruptive thoughts, increasing self-confidence, perception, and positive attitude to perform the sport (dispositional flow) [12]. Cherup and Vidic (2019), also with a longitudinal design study made up of gymnasts, applied a combined program of Mindfulness-Acceptance-Commitment (MAC) and MSPE and found improvements in sports flow indicators, but not in stress [13]. In the above-mentioned works, there was no control group; this fact prevents assessing the impact of mindfulness programs since each athlete begins with different stress levels. Thus, the importance of controlled studies that corroborate the favorable effects of mindfulness on sports stress is necessary.

The evaluation methods of these psychological interventions are sophisticated, and their assessment has been preferably qualitative [14]. In this sense, Kellmann and Kallus (2001) [15] proposed the Recovery-Stress Questionnaire Sport (RESTQ-76 Sport) to psychometrically analyze the stress–recovery balance in athletes of various sports. This instrument has been transcribed into Spanish and validated in a Spanish-speaking population [16], and is currently applied to Mexican soccer players, presenting high reliability in its different dimensions ($\alpha{\sim}0.89$) [17]. Further, various studies have shown that this instrument presents adequate sensitivity to measure the stress–recovery balance during different stages of training and competition [18,19].

Heart rate is regulated by the sympathetic (norepinephrine) and parasympathetic (acetylcholine) components of the autonomic nervous system. In this sense, heart rate variability (HRV: variation over time of the period between consecutive heartbeats) has been shown to be a sensitive marker of the stress–recovery balance [1,20,21]. Any physical or mental imbalance induces stress to maintain homeostasis, and a reduction in stress favors the sympathetic–vagal balance by reducing sympathetic activity, increasing HRV. Instead, stress increases sympathetic activity and decreases HRV [1,2,7,22]. In this sense, it has been reported that mindfulness increases parasympathetic activity and HRV in healthy people [23].

We did not find studies where mindfulness is applied as a psychological intervention to maintain the stress–recovery balance in professional soccer players during the competitive stage, and at the same time, assess its psychometric and physiological effects. This work examines the effect of six weeks of MSPE on the stress–recovery balance in professional soccer players during a competitive season, using RESTQ-76 Sport and HRV as psychometric and physiological evaluation methods. We expect the intervention group (MSPE) to improve the stress–recovery balance and HRV-like physiological stress marker compared to the control group.

## 2. Materials and Methods

### 2.1. Participants and Bioethical Principles

Before the study, the protocol was explained to the participants (n = 52), coaches, and parents; further, they were asked to sign the informed consent and/or assent letter, according to the bioethical principles of research in human beings [24]. The participants freely decided to enter and there was prejudice to those who decided not to participate. Both teams were players whose sociodemographic status (middle class students), physical characteristics, and sports and competitive activities were

similar. The weeks' activities consisted of training for 2 h from Monday to Friday, games on weekends, and breaks the day after the game. Similarly, both teams had nine games at home and out of the city. Given the indicated similarity between both teams, it allowed us to compare them with each other. Injured athletes (4 controls and 1 experimental), those who were taking drugs (1 experimental), and those who did not comply with at least 80% of the treatment sessions (4 experimental) were excluded. The final sample consisted of 42 male athletes from 2 professional third-division soccer teams. Due to the facilities to carry out the mindfulness sessions, each team decided which group they participated in: control without treatment (n = 22, 17.4 ± 1.3 years, height 173.3 ± 7.2 cm, and bodyweight 62.9 ± 7.0 kg) or experimental (n = 20, 16.9 ± 1.3 years, height 171.0 ± 5.7 cm, and bodyweight 62.0 ± 9.1 kg). Before the studies, the protocol was approved by the Bioethics Committee of the Autonomous University of Ciudad Juarez: No. CIP-ICB-2018-1-04.

### 2.2. Experimental Design

MSPE was applied from Monday to Friday for six consecutive weeks, starting at the first opening week of sports competitions. The evaluations (application of the RESTQ-76 Sport questionnaire and registration of the HRV) were carried out one day before the competitions, at the third, sixth, and eighth weeks (two weeks after treatment finished). Before any evaluation, the athletes were asked to refrain from intense physical exercise, smoking, and consuming coffee, tea, alcohol, or energy drinks for at least 24 h.

### 2.3. Treatment: Application of the MSPE

MSPE was applied as a group form at the end of each training session (19–20 h). A trained specialist psychologist made this application according to an already published methodology [8], briefly described here.

- Week 1. Building the base of mindfulness: duration 15–20 min.
    - Activities: Savoring a sweet, diaphragmatic breathing, and breathing-centered meditation;
- Week 2. Strengthen attention: duration 20–30 min.
    - Activities: Body scan and breathing centered meditation;
- Week 3. Stretch the limits of the conscious body: duration 30–45 min.
    - Activities: Meditation focused on the body as a whole and yoga postures (mindful yoga);
- Week 4. Embrace what is calm: duration 30–45 min.
    - Activities: Mindful yoga and walking meditation;
- Week 5. Conscious sports practice: duration 20–40 min.
    - Activities: This week, the participant was asked to use, during their sports practice, what they learned during weeks 1 to 4, paying particular attention to their alertness, breathing, and relaxation;
- Week 6. End and beginning: duration 30–45 min.
    - Activities: All of the above activities.

### 2.4. Psychometric Determination of the Stress–Recovery Balance

For this determination, the adapted and validated (internal validity and reliability) Spanish version of RESTQ-76 Sport [15,16] was applied. This questionnaire consists of 76 items, each one with seven response options on a Likert-type scale, ranging from 0 (never) to 6 (always), where the higher the score, the greater the stress or the recovery, e.g., In the last (3) days/nights I watched TV . . . very often (very often has a value of 4). The RESTQ-76 Sport is divided into 19 scales grouped into four dimensions (I-IV): I. non-sport-specific stress (NSSS), scales 1–7, II. non-sport-specific recovery (NSSR), scales 8–12, III. sport-specific stress (SSS), scales 13–15, and IV. sport-specific recovery (SSR), scales 16–19. These four dimensions are grouped into two integrals groups (V, VI): V. global stress (GS: NSSS + SSS), and VI. global recovery (GR: NSSR + SSR). Finally, a global index (VII) is obtained, also called the stress–recovery balance (GI: GR-GS). For more detail on its items, scales, dimensions, validation, and methodology, consult elsewhere [15,16].

### 2.5. Physiological Determination of the Stress–Recovery Balance

For this determination, the HRV was recorded 48 h after the sports competition, using POLAR TEAM 2® heart rate monitors (Polar, Finland). The records were made in a rest state and outdoors under a shed. The HRV measurement was made for ten continuous minutes in a sitting position, after 5 min of resting, and before the training session, with the methodology elsewhere described [15,16]. The data were processed with the Kubios program version 2.2 (University of Eastern Finland, Kuopio, Finland). This manuscript only reports the nonlinear parameters of the Poincaré diagram called SD1 (changes in parasympathetic activity) and SD2 (inverse of sympathetic activity). SD2 was used to obtain the indicator of sympathetic activity stress score (SS = $1000 \times 1$/SD2). SD1 together with SS served to obtain the sympathetic/parasympathetic index (SS/SD1), also called S:PS [14]. Values of SS and S:PS higher than 10 and 0.3, respectively, are indicators of stress and high sympathetic activity [21].

### 2.6. Statistical Analysis

The statistical analysis was carried out by a researcher who was not involved in the protocol procedures and who did not know the sample characteristics. The Shapiro–Wilk test was applied to analyze the data distribution in each group, and the Levene test for homoscedasticity. Due to the violations in the mentioned parametric assumptions, the variables were examined by non-parametric analyses. The Mann–Whitney U test was used to analyze differences between groups for each time (baseline, half, end, and two weeks of follow-up). The differences between time were analyzed by the Friedman test for multiple comparisons, and by the Wilcoxon test for pairwise comparisons. Finally, because of differences between groups in basal conditions, the MSPE effect on the studied variables was analyzed by delta ($\Delta$) values, subtracting the baseline value to each time. The partial eta-square ($\eta^2$) by two-way ANOVA was used to report the effect size [25]. In the tables, the values are presented as means (X) ± standard deviation (SD) and in the figures as means ± standard error of the mean (SEM). For statistical significance, a value of $p < 0.05$ was considered. The data were analyzed with IBM© SPSS© Statistics version 25.0 software.

## 3. Results

### 3.1. Effect of MSPE on the Stress–Recovery Balance

The effect of MSPE on the stress–recovery balance measured by the RESTQ-Sport 76 questionnaire is reported in Table 1 (absolute values) and Figure 1 ($\Delta$ values). It is noteworthy that, in basal conditions, the experimental vs. the control group showed higher SSS (~0.53 units, $p < 0.05$), higher NSSS (~0.48 units, $p < 0.05$), and higher GE (~0.44 units, $p < 0.05$), and therefore less stress–recovery balance (~−0.98 units, $p < 0.05$). These differences were maintained until the middle of the MSPE program and eliminated at the end and two weeks of follow-up. When basal conditions were eliminated, and both groups compared again, the GE decreased in the experimental group (~−0.50 and ~−0.82 units at the end and two weeks of follow-up, respectively, $p < 0.05$), the GR increased (~0.75 and ~0.71 units at the end and two weeks of follow-up, respectively, $p < 0.05$), and the stress–recovery balance increased (~1.24 and ~1.54 units at the end and two weeks of follow-up, respectively, $p < 0.01$). The longitudinal analysis tells us that, from the end of the treatment to two weeks of follow-up, the experimental group decreased stress and increased recovery in most of the RESTQ-Sport 76 questionnaire dimensions: lower GE (−0.35 ± 0.62 and −0.56 ± 0.62 units for the end and two weeks of follow-up, respectively, $p < 0.01$), higher GR (0.54 ± 0.98 and 0.65 ± 0.59 units for the end and two weeks of follow-up, respectively, $p < 0.01$), and higher stress–recovery balance (0.89 ± 1.40 and 1.20 ± 1.03 units for final and two weeks of follow-up, respectively, $p < 0.01$). The effect sizes were large for the differences between groups and time: GE = 0.19 and 0.13; GR = 0.19 and 0.30; and SRB = 0.30 and 0.12, for differences between groups and time, respectively (Table S2; Supplementary Materials).

**Table 1.** Changes (absolute values) in the perception of the stress–recovery balance during six weeks of mindfulness in third-division professional soccer players, analyzed by the RESTQ-76 Sport questionnaire.

| | Basal, a | | Middle of the Treatment, b | | Final, c | | Two Weeks of Follow-up, d | |
|---|---|---|---|---|---|---|---|---|
| | Control | Experimental | Control | Experimental | Control | Experimental | Control | Experimental |
| I. Non-sport-specific stress | 1.54 ± 0.76 | 2.07 ± 0.80 * | 1.41 ± 0.66 | 2.21 ± 0.95 * | 1.73 ± 1.69 | 1.69 ± 0.42 [b] | 1.83 ± 0.81 | 1.45 ± 0.44 [*,a,b] |
| 1.General stress | 1.11 ± 1.03 | 1.53 ± 0.96 | 0.80 ± 0.79 | 1.94 ± 1.20 * | 1.17 ± 0.96 | 1.41 ± 0.52 | 1.53 ± 1.12 [b] | 1.45 ± 0.71 |
| 2.Emocional stress | 1.51 ± 0.77 | 1.89 ± 1.20 | 1.19 ± 0.81 | 2.08 ± 1.20 * | 1.75 ± 1.03 | 1.55 ± 0.63 | 1.66 ± 0.99 | 1.36 ± 0.44 |
| 3. Social stress | 1.25 ± 1.12 | 1.91 ± 1.00 * | 1.10 ± 0.98 | 2.01 ± 1.10 * | 1.28 ± 1.10 | 1.39 ± 0.47 | 1.91 ± 1.21 [b] | 1.35 ± 0.40 [a] |
| 4. Conflict/Pressure | 2.18 ± 0.73 | 2.65 ± 0.79 * | 2.40 ± 0.83 | 2.50 ± 0.81 | 2.32 ± 0.76 | 2.25 ± 0.69 | 2.36 ± 0.95 | 1.79 ± 0.61 [*,a,b] |
| 5. Fatigue | 1.59 ± 0.89 | 2.09 ± 1.11 | 1.34 ± 0.67 | 2.11 ± 0.98 * | 1.42 ± 0.97 | 1.84 ± 0.61 | 1.59 ± 0.97 | 1.30 ± 0.49 [a,b,c] |
| 6. Lack of energy | 1.70 ± 0.89 | 2.04 ± 1.08 | 1.48 ± 0.77 | 2.49 ± 0.96 * | 2.02 ± 0.79 | 1.96 ± 0.54 | 1.88 ± 0.65 | 1.45 ± 0.48 [*,b,c] |
| 7. Physical disturbances | 1.32 ± 0.90 | 1.61 ± 1.29 | 0.97 ± 0.55 | 1.94 ± 0.93 * | 1.51 ± 0.97 | 1.43 ± 0.74 | 1.42 ± 0.91 | 1.24 ± 0.62 [b] |
| II. Non-sport-specific recovery | 4.09 ± 0.91 | 3.67 ± 1.15 | 4.38 ± 0.62 | 3.86 ± 0.96 * | 4.16 ± 0.96 | 4.07 ± 0.51 | 4.16 ± 0.13 [a,b,c] | 4.16 ± 0.13 [a,b,c] |
| 8. Success | 3.83 ± 0.82 | 3.26 ± 1.19 | 4.01 ± 0.64 | 3.50 ± 1.00 * | 3.95 ± 1.14 | 3.53 ± 0.69 | 3.65 ± 0.89 | 4.28 ± 0.61 [*,a,b,c] |
| 9. Social recovery | 4.18 ± 1.23 | 3.80 ± 1.49 | 4.51 ± 1.02 | 3.90 ± 1.03 | 4.17 ± 1.35 | 4.15 ± 0.82 | 4.11 ± 1.26 | 4.35 ± 0.75 |
| 10. Physical recovery | 3.81 ± 1.24 | 3.23 ± 1.12 | 4.10 ± 0.82 | 3.39 ± 1.06 * | 3.81 ± 1.29 | 3.84 ± 0.62 | 3.91 ± 0.92 | 4.06 ± 0.72 [a,b] |
| 11. General well-being | 4.40 ± 1.38 | 4.06 ± 1.35 | 4.83 ± 0.95 | 4.16 ± 1.02 * | 4.48 ± 1.14 | 4.65 ± 0.72 | 4.31 ± 1.17 | 4.41 ± 0.83 |
| 12. Sleep quality | 4.01 ± 1.23 | 3.40 ± 1.11 | 4.44 ± 0.89 | 3.70 ± 0.91 * | 4.13 ± 1.07 | 4.05 ± 0.66 | 4.11 ± 0.98 | 4.34 ± 0.62 [a,b] |
| III. Sport-specific stress | 1.67 ± 1.16 | 2.15 ± 0.76* | 1.42 ± 0.96 | 2.70 ± 1.15 * | 1.89 ± 0.89 | 1.65 ± 0.39 [b] | 1.83 ± 1.28 | 1.72 ± 0.51 [b] |
| 13. Altered rest periods | 1.53 ± 1.31 | 1.90 ± 1.00 | 1.36 ± 1.23 | 2.51 ± 1.28 * | 1.63 ± 0.93 | 1.60 ± 0.60 [b] | 1.82 ± 1.28 | 1.79 ± 0.46 |
| 14. Burnout | 1.32 ± 1.14 | 2.35 ± 1.25 * | 1.18 ± 0.93 | 2.64 ± 1.31* | 1.65 ± 0.93 [b] | 1.83 ± 0.67 [b] | 1.74 ± 1.16 | 1.45 ± 0.60 [a,b] |
| 15. Physical injuries | 1.75 ± 1.08 | 2.18 ± 1.19 | 1.49 ± 0.94 | 2.60 ± 1.07 * | 1.91 ± 0.85 | 1.74 ± 0.47 [b] | 1.91 ± 1.38 | 1.46 ± 0.51 [b] |
| IV. Sport-specific recovery | 4.02 ± 1.03 | 3.76 ± 1.26 | 4.52 ± 0.99 | 3.82 ± 0.63 * | 3.58 ± 1.08 [b] | 4.33 ± 0.95 * | 3.91 ± 0.77 [b] | 4.25 ± 0.77 [a] |
| 16. Well-being/Fitness | 4.11 ± 1.24 | 3.81 ± 1.31 | 4.60 ± 0.79 | 3.76 ± 0.92 * | 3.58 ± 1.39 [b] | 4.30 ± 0.89 | 4.01 ± 0.93 [b] | 4.33 ± 0.67 [b] |
| 17. Personal fulfillment | 3.41 ± 0.85 | 3.26 ± 1.06 | 3.84 ± 1.05 | 3.41 ± 1.01 | 3.05 ± 0.91 | 3.93 ± 0.52* | 5.58 ± 0.96 | 4.01 ± 0.72 [a] |
| 18. Self-efficacy | 4.23 ± 1.14 | 3.79 ± 1.28 | 4.57 ± 0.95 | 3.75 ± 1.07 * | 3.74 ± 1.18 [b] | 4.33 ± 0.80 | 3.88 ± 0.85 [b] | 4.16 ± 0.83 |
| 19. Self-regulation | 4.01 ± 1.40 | 3.79 ± 1.24 | 4.68 ± 0.91 [a] | 4.19 ± 0.95 | 3.59 ± 1.33 [b] | 4.48 ± 0.80 * | 3.85 ± 0.96 [b] | 4.36 ± 0.65 [*,a] |
| V. Global Stress | 1.63 ± 0.98 | 2.07 ± 0.56 * | 1.38 ± 0.76 | 2.48 ± 0.95* | 1.77 ± 0.70 | 1.69 ± 0.36 [a,b] | 1.82 ± 0.98 | 1.59 ± 0.42 [a,b] |
| VI. Global recuperation | 4.04 ± 0.93 | 3.72 ± 1.08 | 4.47 ± 0.83 [a] | 3.81 ± 0.60 * | 3.91 ± 0.95 [b] | 4.15 ± 0.51 [a,b] | 3.99 ± 0.73 [b] | 4.30 ± 0.77 [a,b] |
| VII. Global index | 2.62 ± 1.76 | 1.64 ± 1.14 * | 3.21 ± 1.59 [a] | 1.49 ± 1.06 * | 2.29 ± 1.46 [b] | 2.44 ± 0.69 [a,b] | 2.26 ± 1.40 [b] | 2.84 ± 1.31 [a,b] |

Data on a Likert-type scale: ranging from 0 (never) to 6 (always). Roman numerals represent dimensions, and Arabic numerals represent scales within each dimension. The values represent means ± SD, the asterisks (*) differences between groups, and the indices (a, b, c) basal, half of the treatment, and final differences between time. $p < 0.05$.

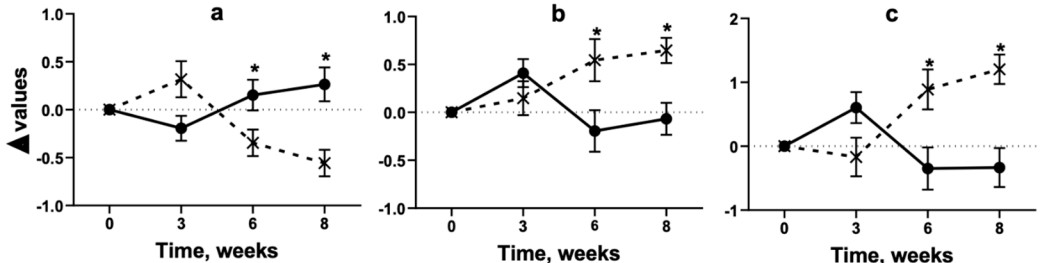

**Figure 1.** Changes (Δ values) in the perception of the stress–recovery balance during six weeks of mindfulness in third-division professional soccer players, analyzed by the RESTQ-76 Sport questionnaire. (**a**) Global stress, (**b**) global recovery, and (**c**) stress–recovery balance. Values represent mean ± SEM. Solid lines = control group, dotted lines = experimental group. Asterisks represent statistical differences between groups, $p < 0.05$.

### 3.2. Effect of MSPE on HRV

The stress–recovery balance values measured by the nonlinear indicators of HRV (SD1, SD2, SS, and S:PS) were not modified by the MSPE program (absolute values in Table 2; Δ values in Figure 2).

**Table 2.** Changes (absolute values) in heart rate variability during six weeks of mindfulness in third-division professional soccer players.

| | Basal | | Middle of the Treatment | | Final | | Two Weeks of Follow-up | |
|---|---|---|---|---|---|---|---|---|
| | Control | Experimental | Control | Experimental | Control | Experimental | Control | Experimental |
| SD1 | 28.23 ± 15.86 | 29.46 ± 9.76 | 24.06 ± 14.32 | 31.65 ± 16.03 | 35.40 ± 24.05 | 35.17 ± 17.22 | 27.31 ± 16.41 | 32.84 ± 15.54 |
| SD2 | 105.36 ± 39.71 | 107.59 ± 27.98 | 96.37 ± 37.85 | 113.50 ± 34.88 | 113.77 ± 56.65 | 120.83 ± 43 | 112.29 ± 40.79 | 125.52 ± 47.65 |
| SS | 10.61 ± 3.36 | 9.95 ± 2.76 | 11.95 ± 4.91 | 9.75 ± 3.40 | 10.40 ± 3.93 | 9.24 ± 2.93 | 10.15 ± 3.96 | 8.83 ± 2.69 |
| S:PS | 0.53 ± 0.36 | 0.41 ± 0.29 | 0.72 ± 0.63 | 0.45 ± 0.39 | 0.47 ± 0.50 | 0.34 ± 0.20 | 0.65 ± 0.76 | 0.53 ± 0.98 |

SDI = parasympathetic activity, SD2 = inverse of sympathetic activity, SS = stress score, S:PS = sympathetic/parasympathetic index. Values are means ± SD.

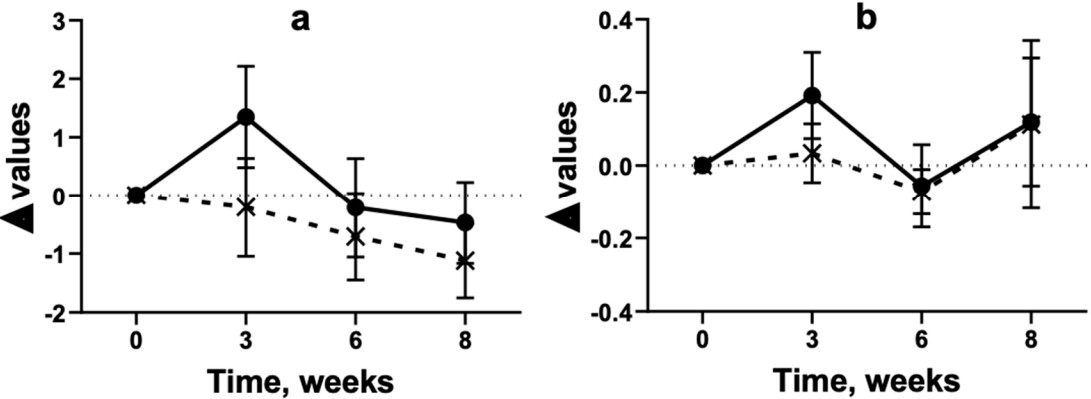

**Figure 2.** Changes (Δ values) in heart rate variability during six weeks of mindfulness in third-division professional soccer players. (**a**) Stress score (SS), (**b**) index of sympathetic/parasympathetic activity (S:PS). Values represent mean ± SEM. Solid lines = control group, dotted lines = experimental group.

## 4. Discussion

The present is a longitudinal case–control study to analyze whether six weeks of the MSPE program modified the perception of the stress–recovery balance, the stress score indices, and the relation of sympathetic–parasympathetic activity of the autonomic nervous system, in two teams of professional soccer players during a competitive season.

The main results show that six weeks of MSPE increased the stress–recovery balance measured by the RESTQ-Sport 76 psychometric questionnaire and overall recovery ($p$ <0.5), but decreased overall stress. This effect is observed from three weeks after treatment begins, remaining until two weeks of follow-up. The control group's changes were not statistically significant; it is noteworthy that only halfway changes through the treatment were observed and subsequently disappeared when the absolute values were analyzed. However, when the basal effect was eliminated ($\Delta$ values), the differences appeared from the end until two weeks of follow-up, and also the effect size ($\eta^2$) in all found differences was larger (0.12–0.30) [25]. The MSPE program did not modify HRV.

De Petrillo (2009), studying long-distance running athletes by an MSPE program, reported a decrease in the indicators of anxiety, concern for perfection, and disruptive thoughts, whilst on the other hand, an increase in the enjoyment of sports [26]. Dehghani et al. (2018), studying university athletes from various disciplines, and who received an MAC program, reported a decrease in sports and cognitive anxiety, and an increase in sports performance and self-confidence [27]. Mehrsafar et al. (2019), in a case–control study with Wuhsu athletes under a mindfulness program that combined mindfulness-based stress reduction (MBSR), MAC, and mindfulness-based cognitive therapy (MBCT), reported a decrease in somatic and cognitive anxiety and an increase self-confidence [28]. Our MSPE program did not modify the perception of social recovery and personal fulfillment, these two variables have been reported to be less affected by sport stress in young elite soccer players [18]. Similar to our study, Mehrsafar et al. (2019) addressed a little aspect cited in the literature: impact of mindfulness on the athlete's recovery phase within a training and competition program, and its post-treatment effects [28]. De Petrillo et al. (2009) and Mehrsafar et al. (2019) also found positive effects of the mindfulness program for up to four and eight weeks after treatment [26,28]. However, randomized controlled trials are required to recommend mindfulness as a stress–recovery treatment and assess its medium- and long-term effects, especially with MSPE.

A factor that could favorably influence the observed result on the stress–recovery balance in the experimental group was the weekly informative talk given to the athletes by the instructor when changing each activity of the MSPE program. Here, the athletes externalized their perceived experiences during the practice of MSPE, and the instructor gave their technical explanations about it, with recommendations proposed by Kellmann and Kallus in the MSPE manual [15]. However, placebo effects have been reported in mindfulness studies [29]. For that, we consider that to rule out the placebo effect of MSPE, more controlled studies and specific physiological measurements such as HRV must be jointly analyzed with measurements made here.

As indicated in the methods, here was used the Spanish and validated version by Gonzalez-Boto et al. [16]. In the same way, to support the above results, with the 42 participants' data, the reliability of the RESTQ-Sport 76 questionnaire was analyzed (Table S1; Supplementary Materials), finding high Cronbach's alpha values: 0.89, 0.77, 0.88, 0.91, 0.91, 0.93, and 0.82, for non-sport-specific stress, non-sport-specific recovery, sport-specific stress, sport-specific recovery, global stress, global recuperation, and global index, respectively; these results are similar to those reported by Lope-Fernández and Solis-Briceño (2020) in a small sample (n = 22) of Mexican soccer players [17]. In other words, the RESTQ-Sport 76 presents high reliability to evaluate recovery stress in soccer players. However, larger sample studies are necessary to verify its reliability and validity in Mexican athletes.

HRV is considered as a sensitive indicator to assess the athlete's stress–recovery status [1,19,21]; it was analyzed using the nonlinear indicators SD1, SD2, SS, and S:PS ratio—the last two recently being proposed as ideal sports stress indexes [21]. Two indicators (SS and S:PS) were analyzed from the Poincaré graph with data from professional soccer players during 11 months of the competitive season, being both reliable indicators, since they are directly proportional to the athlete's stress state. Although, to date, various criteria have been proposed to interpret HRV, these should be supported by more studies to corroborate as health indicators in athletes. In this context, the experimental group maintained average SS values considered as an alert of stress [8–10], while the control group exceeded this value, considered as a high level of sympathetic stress [19]. On the other hand, according to

Naranjo et al. (2015) [21], both groups presented high values in the S:PS ratio, indicating a high presence of stress; the experimental group had slightly lower values, mainly when the data are analyzed in absolute manner. Despite the trends mentioned above, none of the SS and S:PS averages were statistically different, either in time or between groups. HRV is affected by various aspects such as age, temperature, blood pressure, and body position, among others [30]. In the present study, the measurement of HRV was performed in a sitting posture as proposed by Naranjo et al. (2015) [21]; however, to reduce muscle contractions' noise in the HR record, the most appropriate resting posture was to lie supine [31]. Further, HRV was evaluated in the afternoons in an outdoor space, therefore, under uncontrolled temperature. The preceding is vital to mention since, on the dates of the study (September–November), the average environmental temperature decreased (from 24 to 11 °C). Thus, before attributing the absence or presence of any impact of MSPE on HRV, it should be considered that a lack of posture and temperature control generated biases in this variable, preventing statistical differences to be detected in the observed trends.

Although the SS and S:PS ratio indicators have been recently created, they have been applied in several studies with athletes. Demarzo et al. (2015) reported an increase in parasympathetic activity in the first two weeks of regular mindfulness practice, which was interrupted by the athletes' competitions participation [32]. On the other hand, according to Miranda-Mendoza et al. (2018), working with a handball team during a competitive season and without any psychological treatment [33], their SS values rose during the competition and returned to an initial level 72 h after the competition ended (final: $14.08 \pm 4.87$; 72 h: $8.89 \pm 4.34$).

## 5. Limitations

The limitation of this study was the non-randomization of both groups, due to facilities for assessments and mindfulness practice.

## 6. Conclusions

Six weeks of MSPE improves the stress–recovery balance, measured by the psychometric questionnaire RESTQ-Sport 76 in third-division professional soccer players during the competitive season, reduces stress, and increases recovery. A positive effect was not observed in the nonlinear indicators of the VFC: SD1, SD2, SS, and S:PS. It is proposed to carry out future randomized controlled trials that ensure the findings found here, analyzing diverse populations and sport disciplines, in both sexes. Given the inherent subjectivity of psychometric questionnaires, such studies must be accompanied by physical, biochemical, and physiological measurements.

**Supplementary Materials:** The following are available online at http://www.mdpi.com/2071-1050/12/17/7091/s1.

**Author Contributions:** J.H.-R. was responsible for the recruitment of participants, conducted the MSEP program, and led the development of this manuscript. A.R.-J. and R.P.H.-T. wrote the manuscript, designed the study procedures, and performed the statistical analyses of the data. J.T.Q.-C. conducted the physical exercise program. M.S.C.-B. designed the study. All authors have read and agreed to the published version of the manuscript.

**Funding:** This research was funded by Programa para el Desarrollo Profesional Docente en Educación Superior (PRODEP).

**Acknowledgments:** J.H.-R. was supported by a Ph.D. scholarship from the Consejo Nacional de Ciencia y Tecnología (CONACyT) No. 635259. The authors acknowledge Oscar Armando Esparza del Villar and Luis Felipe Reynoso Sánchez for academic support at graduate student J.H.-R, and Rafael Villalobos Molina for style corrections and English.

**Conflicts of Interest:** The authors declare no conflict of interest.

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
