# Peer review of "Effect of Mindfulness on the Stress–Recovery Balance in Professional Soccer Players during the Competitive Season"

_sustainability, doi:10.3390/su12177091_

Round 1

Reviewer 1 Report

Thanks to the authors for submitting this interesting paper, which describes and evaluates a mindfulness intervention with two soccer teams. I feel that the paper adds something to the picture of mindfulness-based interventions conducted in sport settings worldwide. I have a few comments that I hope could help to improve the final contribution of the paper.

Overall, I think the authors have done a good job. One strength of the study is that stress was recorded with both questionnaires and physiological markers. My main concerns relate to grouping, data analysis and interpretation of the data

Please excuse grammatical and spelling errors in this review, English is not my first language. Having said this, I cannot rate the quality of the English language of the paper. I suggest that the authors have the paper proofread by an English native.

Introduction

Lines 33ff.: I feel “attacked” is a very strong word and the word “pressures” does not describe the mentioned aspects accurately. I suggest to use something like “professional athletes have to deal with various forms of sport-specific stressors, such as….”. Also, I suggest to also mention the high training load as a central feature of an athletes’ life. Overtraining (as mentioned by the authors) would not occur without a lot of training.

Line 40.: I suggest to not speak of “psychotherapeutic” techniques but of “psychological interventions”. In my opinion, psychotherapy describes a therapeutic treatment and MSPE does not fall under it.

Line 48ff: I would not focus on the questionnaire here (the RESTQ), but on the concept, namely stress-recovery balance. In my opinion, this section should focus on the potential impact of mindfulness interventions on the stress-recovery balance, rather than on how stress-recovery balance is assessed. The questionnaire is a means to grasp the concept, it is sufficient if the questionnaire is first mentioned and described in the methods section.

Line 63: Please state your hypotheses at the end of the introduction. Something like “We expect the intervention group (MSPE) to improve stress-recovery balance and physiological stress markers (HRV) compared to the control group”

Method

Line 66: please indicate what kind of briefing you talk about

Line 70: Please indicate whether the excluded subjects differed from the rest of the intervention group in terms of demographic data or stress parameters.

Line 71ff: It would be important to explain here in more detail how the grouping came about. Were the athletes in each team free to make their own decisions and was there an intervention group in each of the two teams? Or did the team as a whole have to decide? Could both teams have decided to participate in the intervention? What happened to the players who would have (not) wanted to participate but were outvoted by the majority of the team? Is there a reason why no randomized grouping was made? These points are central for the interpretation of the results. Please also indicate here whether the two groups differed in terms of demographic data.

Line 108: I suggest to report Cronbach Alpha values of the scales at baseline (maybe it is possible to integrate this information in Table 1)

Line 128:… and the Levene test to check for homoscedasticity

Results

Lines 142ff: The differences between the groups at baseline support my previous comments on grouping. It seems that athletes who have a worse stress-recovery balance are more likely to choose to participate in the intervention. The improvements in stress parameters in the intervention group may therefore be due to regression to the mean. It is therefore somewhat misleading, in my opinion, to equate the baseline of the two groups and to assess changes from there as it is easier to improve if the balance is poor. I would therefore suggest to report the absolute values and check whether the stress-recovery balance of the intervention group balances out over time.

Discussion

Line 184ff: I suggest to move the description of how much the teams train and play to the methods section

Line 188ff: I would be more cautious in interpreting the results due to the above mentioned points regarding grouping and differences at the beginning of the intervention. It seems that the participants of the intervention group rebalance their stress-recovery balance over time, this could be due to the intervention or due to regression to the mean.

Lines 196ff: In this section the current state of knowledge is described and gaps in research are pointed out, this usually happens in the introduction. I recommend that the authors consider moving this section to the Introduction.

Lines 220ff: The same applies to this section as in the previous commentary. This section explains why MSPE favors recovery of athletes, which would be better addressed in the introduction.

Lines 225ff: Here the authors make an important point. Since it was a waiting list control group, it is not possible to judge which aspects of the intervention are relevant. Was it mindfulness? Was it the posibility to discuss things in a group? Was it a placebo effect?

I recommend that the authors include a section in which they discuss the limitations of the study. Some aspects I have already mentioned. Another limitation, in my opinion, is that no manipulation check was performed for the intervention. It would be important to know whether mindfulness interventions actually lead to more mindfulness.

I propose that the authors revise the conclusion again after the adjustments in the article.

Reviewer 2 Report

First: In General: it's a good paper Title: the title properly explain the purpose and objective of the article> Abstract: abstract contains an appropriate summary for the article, language used in the abstract easy to read and understand, there are no suggestions for improvement. Introduction: authors do provide adequate background on the topic and reason for this article and describe what the authors hoped to achieve. Results: the results presented in a clear manner, the authors provide accurate research results, there is sufficient evidence for each result. Conclusion: in general: Good and the research provides ample data for the authors to make their conclusion. Grammar: Need Some revision.

Reviewer 3 Report

This is an interesting work with a longitudinal approach assessing the effects of the Mindful Sports Performance Enhancement effect on stress-recovery.

The authors may carefully review the English of this manuscript. It is possible to find some Latin origin sentences typically used by Spanish, Portuguese and Italians. A Latin origin reader may easily understand the manuscript, however Anglo-Saxon reader may not.

This work present a set of issues that the authors may assess before publication. First, I am wondering if and how a paper related to sports sciences falls in the scope of Sustainability? 

Abstract: The metrics may be in the Units of International System (meters).

Introduction; L.53: Please explain what HRV is. And why HRV increases reducing stress. Some readers outside the sports sciences may confound it with HR or may not be familiarized with it.

Methods L.113: Do this questionnaire present psychometric characteristics? Was this validated for the sample of this study? The authors argue that this is a validated Spanish version. But were the participants Spanish people? Is this a valid instrument for French, American, Germans or Brazilians, elite or amateur? If yes, please state; if not, please justify the instrument use.

Methods L 116-118: The evaluations were in the the soccer field? Who assessed the HRV? Should not be in a lying down position? What the literature says about the resting period to measure resting HR?

Results and discussion: Why the Social Recovery, personal fulfilment and self-efficacy were not affected? Discuss it.

L. 194: The authors used non-parametric tests. Which test was used to assess effect sizes? Please present the effect sizes data.

L. 196-197: But in the present study Social Recovery, personal fulfilment and self-efficacy were not affected. Social recovery and personal fulfilment were not changed. The authors may discuss why. Those are variables related with life quality...

L220-224: Should this be in introduction or methods section? In the introduction section, the MSPE lacks of information.

L. 225-228: But this factor influenced all the variables? Or some of the variables? Which ones the authors believe that were mostly influenced by this talk?

L. 243: This reviewer have severe doubts if seated position is the most adequate position.

L.246-248: But, why did the authors found no changed in HRV? Was that because of which factors? Is the HRV the most accurate physiological factor? Perhaps not... if so, how to measure physiological stress? In fact, HRV have been used to assess physiological stress (i.e., internal mechanical work) result from external mechanical work... but is this an accurate measure for psychological stress?

Discussion: This study do not have limitations.

Round 2

Reviewer 3 Report

The authors have made a considerd effort to improve the manuscript.